# Fragment Screening in the Development of a Novel Anti-Malarial

Xiaochen Du [†] [ORCID], Ran Zhang [†] and Matthew R. Groves *

Structural Biology in Drug Design, Groningen Research Institute of Pharmacy, University of Groningen, 9713 AV Groningen, The Netherlands; xiaochen.du@rug.nl (X.D.); ran.zhang@rug.nl (R.Z.)
* Correspondence: m.r.groves@rug.nl
[†] These authors contributed equally to this work.

**Abstract:** Fragment-based approaches offer rapid screening of chemical space and have become a mainstay in drug discovery. This manuscript provides a recent example that highlights the initial and intermediate stages involved in the fragment-based discovery of an allosteric inhibitor of the malarial aspartate transcarbamoylase (ATCase), subsequently shown to be a potential novel anti-malarial. The initial availability of high-resolution diffracting crystals allowed the collection of a number of protein fragment complexes, which were then assessed for inhibitory activity in an in vitro assay, and binding was assessed using biophysical techniques. Elaboration of these compounds in cycles of structure-based drug design improved activity and selectivity between the malarial and human ATCases. A key element in this process was the use of multicomponent reaction chemistry as a multicomponent compatible fragment library, which allowed the rapid generation of elaborated compounds, the rapid construction of a large (70 member) chemical library, and thereby efficient exploration of chemical space around the fragment hits. This review article details the steps along the pathway of the development of this library, highlighting potential limitations of the approach and serving as an example of the power of combining multicomponent reaction chemistry with fragment-based approaches.

**Keywords:** X-ray crystallography; drug discovery; fragment-based; protein structure; malaria; *Pf*ATCase

## 1. Introduction

X-ray diffraction provides valuable insights into the atomic and molecular structure of various substances, as it allows us to delve into the microscopic realm and visualize intricate arrangements of atoms or even electrons within a crystal lattice. This technique enables scientists to uncover details that are not perceptible to the naked eye, providing a deeper understanding of protein, peptide, and molecular structures and their interactions [1]. In structural biology, drug discovery, and related areas, X-ray crystallography is used for protein structure determination [2], nucleic acid interactions [3], enzyme mechanism investigation [4], and structure-based drug design and optimization [5]. These developments are also inseparable from the support of related technologies such as sequence identification [6], cloning [7], transformation [8], purification of proteins and peptides [9], and related data processing developments such as XDS [10], HKL2000/3000 [11], SHELX [12], Dials [13], Phenix [14], CCP4 [15], and *PanDDA* (Pan-Dataset Density Analysis) [16].

During data collection, traditionally, the process of mounting and aligning crystal samples for X-ray diffraction experiments was time-consuming since it required manual interventions. However, modern beamlines are equipped with sophisticated robotic systems that can handle sample mounting and alignment with minimal human intervention during data collection and ensure precise and consistent sample handling, which can eliminate human error as much as possible in sample manipulation and positioning and contribute to the reliability of collected data [17]. It is worth noting that remote operation of synchrotron beamlines has been standard for some time, with many beamlines allowing fully automated

data collection [18]. The remote data collection system allows researchers to perform data collection via computer operations at their home laboratory, improving accessibility and efficiency and allowing scientists to control experiments and collect data from their own laboratories. In the field of structural biology research and services, institutions such as EMBL Grenoble [19], EMBL Hamburg [20] and DESY [21], provide expertise and support in structure biology techniques and equipment for crystal data collection. Moreover, these robots can also handle a large number of samples in a high-throughput manner, dramatically speeding up data collection. The development of automated systems can rapidly screen hundreds to thousands of crystallization conditions with a minimum volume of protein, which facilitates the identification of suitable crystallization conditions for high-quality diffraction data [22,23]. Commercial crystallization kits, widely used for systematic screening, can be linked with the high-throughput screening (HTS) method and automated robots to streamline and expedite experiments [24,25]. In addition, the HTS method is also employed in crystal soaking, which helps to optimize soaking conditions to maximize the high quality of crystal structure, including but not limited to screening a range of ligands, concentrations, and soaking times [26]. Automated systems and liquid-handing robots are often used to facilitate the soaking process as well [27,28]. Another critical step in crystal growth is crystal harvesting, where high-throughput crystal harvesting techniques contribute to large-scale crystal fishing and the handling of microcrystals [29].

Since the first structure-based drug, Saquinavir, which was developed as an HIV protease inhibitor using the three-dimensional structure of the HIV protease enzyme [30,31], was approved for the market, this approach has marked the beginning of a new chapter in this field and has revolutionized the way scientists approach the development of new therapeutics. Structure-based drug design has become an integral part of the drug discovery process. It has facilitated the development of numerous successful drugs across various therapeutic areas, including cancer (e.g., Imatinib, mainly targeting chronic myeloid leukemia) [32,33], infectious diseases (e.g., Oseltamivir) [34,35] cardiovascular disorders (e.g., Captopril, mainly targeting the renin-angiotensin-aldosterone system) [36], and neurological conditions (e.g., acetylcholinesterase, for the treatment of Alzheimer's disease) [37,38]. This approach continues to evolve with advancements in computational modeling [39], fragment-based screening [40], and high-throughput screening technologies (HTS) [41,42], which further enhance the efficiency and effectiveness of drug discovery.

Fragment-based screening through X-ray aims to identify weak-affinity interactions between proteins and ligand precursors, which are then designed and optimized through chemical modification, leading to the development of effective drug candidates [40]. In this manuscript, we present the example of a series of inhibitors designed against the aspartate transcarbamoylase of *Plasmodium falciparum* (*Pf*ATCase). The design is based on both a 140-member library X-ray diffraction-based structural screening of an in-house library and a cocktail-based screening of an additional 1020-member in-house fragment library. In an enzymatic assay screening, the crystal data were collected at DESY beamline P11 [43]. A key aspect of the library choice was its suitability for further elaboration by multicomponent reaction chemistry (see below). This study aims to develop antimalarial drug candidates and provides both an introduction to structure-based drug design and optimization, from initial hits to potential lead molecules. Additionally, it demonstrates the value of multi-component reaction chemistry as a mechanism to rapidly elaborate fragments into potent inhibitors. This series of inhibitors resulting from this effort is known as the BDA series, after the initials of the scientist responsible for its creation.

Multi-component reactions (MCRs) are a type of polymerization reaction that involves more than two types of substrates, utilizing a straightforward process [44]. In addition to their advantage of efficient reaction steps, perhaps their more significant advantage is their ability to create diverse starting materials, leading to a variety of scaffolds in the final products [45]. This capability allows for the construction of a specific library with diversity and complexity, which can greatly satisfy the diverse requirements for drug diversity. Furthermore, the concept of multi-component cascade reactions has been introduced [46],

addressing the limitations of the number of MCR reactions and significantly expanding the resulting scaffold diversity.

## 2. Pyrimidine Biosynthesis as a Malarial Target

Malaria remains one of the most life-threatening diseases with a long history, and nearly half of the population in the world is at risk, including infants, children under 5 years old, and pregnant women (https://www.who.int/news-room/fact-sheets/detail /malaria (accessed on 29 March 2023)). It is prevalent in tropical countries and transmitted through the bite of female anopheles mosquitoes, with the causative agent (the most deadly being *P. falciparum*) possessing complex life cycles [47]. According to the World Malaria Report, there were 247 million cases of malaria in 2021, of which more than 80% were children under 5 years of age, with the African region being a high-risk area. Recently, due to factors associated with the outbreak of COVID-19, the number of malaria cases has shown a significant increase (https://www.who.int/news-room/fact-sheets/detail/malar ia (accessed on 29 March 2023)).

The pyrimidine synthesis pathway, salvage pathway, and de novo synthesis pathway play a crucial role in the biosynthesis of nucleotides, which are essential for cellular processes in most organisms [48]. The pyrimidine de novo synthesis pathway contains six distinct enzymatic steps, each catalyzed by specific enzymes: carbamoyl phosphate synthesis (CPS II), aspartate transcarbamoylase (ATCase), dihydroorotase (DHOase), dihydroorotate dehydrogenase (DHODHase), orotate phosphoribosyl transferase (OPRTase), and Orotidine 5'-monophosphate decarboxylase (ODCase). These enzymes convert bicarbonate and glutamine into uridine monophosphate (UMP), with the reaction catalyzed by ATCase recognized as the first committed step (Figure 1) [49]. Since *P. falciparum* lacks uridine-cytidine kinase, deoxycytidine kinase, and thymidine kinase, which are essential for the salvage pathway of pyrimidine nucleosides [50]. The de novo biosynthesis pathway was demonstrated to be the major route for the synthesis of pyrimidine nucleotides, highlighting enzymes within this pathway as potential drug targets. Inhibition of these enzymes can disrupt pyrimidine synthesis, leading to impaired DNA and RNA synthesis and cell growth. Furthermore, an inhibitor known as triazolopyrimidine (DSM256), targeting DHODHase, the fourth step of the de novo pyrimidine synthesis pathway, has demonstrated antimalaria activity in clinical development [51,52]. This development provides further evidence that disrupting de novo pyrimidine synthesis holds promise as a viable approach for malaria treatment. Importantly, the design and optimization of this inhibitor were informed by X-ray structural data, highlighting the significance of structural insights in drug development. In addition to malaria, other diseases such as tuberculosis [53] and trypanosome cruzi [54] may also be addressed through inhibition of the de novo pyrimidine synthesis pathway due to its specific characteristics. Further, the development of novel herbicides may be supported through the investigation of de novo pyrimidine biosynthesis.

**Figure 1.** The de novo pyrimidine biosynthesis pathway. The yellow dotted line shows the Aspartate transcarbamoylase (ATCase) catalytic step.

### 3. The Structure of *Pf* ATCase

Structure-based drug design is an approach that utilizes the three-dimensional structure of a target protein, in this case, *Pf* ATCase, to guide the design and optimization of small-molecule inhibitors. By understanding the binding interactions between the target protein and potential inhibitors, it is possible to develop and examine compounds with a view to improving their selectivity, potency, and kinetic properties. Our research has focused on developing antimalaria drug candidates that aim at inhibiting the de novo pyrimidine biosynthesis pathway, which is crucial for the survival and replication of the malaria parasite. Among the six catalytic enzyme steps, our investigation specifically targets aspartate transcarbamoylase (ATCase) as a potential drug target. First and foremost, obtaining the crystal structure of *Pf* ATCase at high resolution is crucial, both for studying its catalytic mechanism and demonstrating the availability of a suitable system to perform X-ray-based fragment screening. However, it is important to note that obtaining a pure and active sample of the protein is essential both prior to and during the crystallization process. This, coupled with the need to generate high-resolution crystals, remains a major limitation to X-ray-based fragment screening.

Based on the *Pf* ATCase structure (PDB: 5ILQ, 5ILN), it was observed that the *Pf* ATCase protein adopts a trimeric assembly of identical subunits [55]. This trimeric arrangement arranges the three subunits around a central symmetry axis (Figure 2a), with the three

independent active sites of *Pf*ATCase composed of contributions from two different neighboring subunits within the trimeric assembly (Figure 2b). This assembly configuration not only ensures the overall structural integrity of *Pf*ATCase but also raises the potential for cross-talk between molecules during the catalytic cycle. Should such cross-talk exist, the potential for inhibition by an allosteric inhibitor raises the potential for species-selective inhibitors that do not target the conserved active site. In a homologous system (the ATCase of humans; *Hs*ATCase [53]) we have indeed shown that while the classical ATCase inhibitor PALA shows a tri-phasic inhibition of purified *Hs*ATCase, an allosteric inhibitor indeed provides inhibition with a single phase (manuscript submitted).

Each subunit within the trimeric structure of *Pf*ATCase possesses the same fold that contains 11 α-helixes and 9 β-strands [56] and can broadly be divided into a carbamoyl phosphate (CP) and L-aspartate (Asp) binding domain, located at the N-terminal and C-termini of the chain, respectively. Both overall structure and catalytic subunit of *Pf*ATCase demonstrated a high degree of conservation with the well-known *E. coil* ATCase, which serves as a textbook example of ATCase [55,57]. This conservation encompasses not only the sequence but also the secondary structural elements. Furthermore, the active site residues in *Pf*ATCase, such as Ser135 and Lys138 (Figure 2c), align perfectly with the corresponding residues in *E. coil* ATCase, which are known to be an essential part of the catalytic activity of ATCase [55].

The presence of these conserved residues and the overall structural similarities between *Pf*ATCase and *E. coil* ATCase provide valuable insights into the functional aspects of *Pf*ATCase and support the understanding of its catalytic mechanism. This knowledge can aid in the rational design of inhibitors targeting *Pf*ATCase for the development of antimalarial drug candidates.

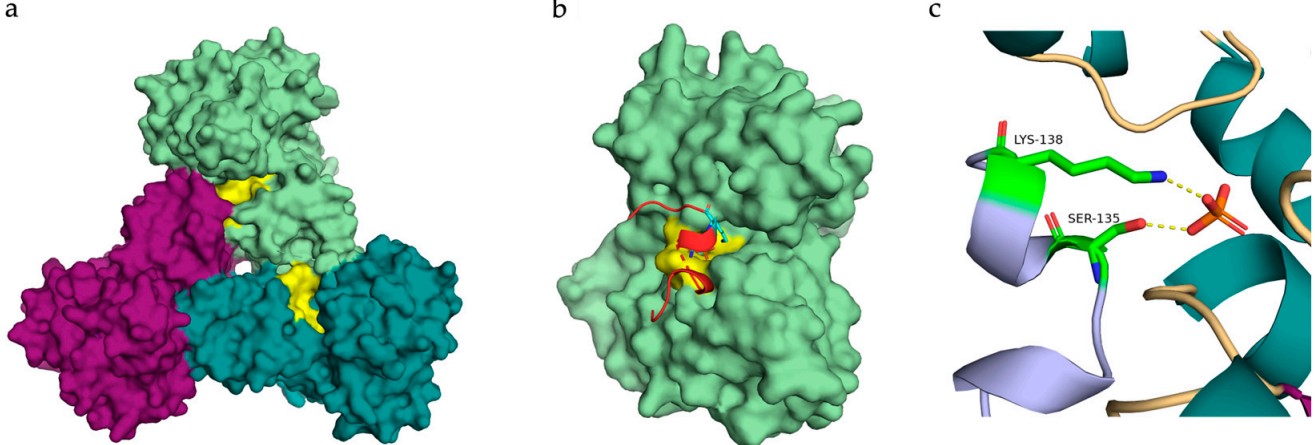

**Figure 2.** (**a**) The trimeric crystal structure of the *Pf*ATCase: citrate complex, which includes three active site pockets (PDB ID: 5ILN), is highlighted in yellow. Each active site pocket is composed of two adjacent protein units. (**b**) Zoomed-in view of the *Pf*ATCase active pocket (highlighted in yellow, PDB ID: 5ILN). Both the yellow-labeled amino acids and the red-labeled 120 s loop of adjacent proteins work together to form the active pocket. (**c**) The Citrate: *Pf*ATCase crystal structure shows Lys138 and Ser135 of the 120 s loop ((highlighted in purple)) demonstrating key interactions with a phosphate ion that mimics a portion of the substrate carbomylphosphate (PDB ID: 5ILN). Figure created using PyMol v.2.5.5 [58].

## 4. The Active Site of *Pf*ATCase

To further confirm the essential active site residues of *Pf*ATCase, our previous work conducted experiments involving the mutation of these residues and performed whole-cell assays [59]. Specifically focusing on a Arg109Ala/Lys138Ala mutant, the structural and functional effects of these mutations and the role of *Pf*ATCase were investigated. Additionally, the crystal structure of the mutant *Pf*ATCase (PDB:6HL7) was determined,

allowing us to observe the specific structure changes. Since the residues Arg109 and Lys138 are located on the interfaces of two subunits, when a mutation occurs in one subunit, it would affect the function, with two out of the three active sites present in the enzyme being impacted. Importantly, the mutation did not significantly disrupt the overall protein assembly, suggesting that the mutated residues did not affect the formation of functional enzyme complexes [59].

Functional assays were followed to assess the enzymatic activity of the mutant *Pf*ATCase, and, through these assays, we were able to confirm the key roles of the active site residues [59]. The results demonstrated that the Arg109Ala/Lys103Ala mutation led to a noticeable decrease or complete loss of catalytic activity, providing further evidence for the essentiality of these residues in the enzyme's function, and subsequent in vivo experiments using this mutant demonstrated the parasite's reliance on correctly functioning *Pf*ATCase.

These experiments, combining structural analysis and functional assays, have provided crucial insights into the importance of the active site residues in *Pf*ATCase and the druggability of *Pf*ATCase. The confirmation of their functional significance also strengthens our understanding of the catalytic mechanism of *Pf*ATCase and paves the way for future studies on the development of inhibitors targeting this enzyme for antimalarial drug discovery.

## 5. Mechanism of *Pf*ATCase

The active site of *E. coil* of ATCase was already reported to have two states, T state and R state [60], with an allosteric site known that allows the parasite to modulate ATCase activity based on UTP and CTP concentrations [61,62]. The apo structure in the T state represents a conformational state with low substrate affinity and low activity. In contrast, the structures in the R state correspond to a conformational state with higher substrate affinity and higher activity. Similar to the ATCase of *E. coil*, *Pf*ATCase was hypothesized to have an allosteric site. Citrate, as an analogue of carbamoyl aspartate (CA), the product of the ATCase catalytic mechanism, was chosen as a probe molecule to perform co-crystallization (PDB: 5ILN), resulting in a model of the R-state *Pf*ATCase (Figure 2). Additionally, comparing the structure of the T-state with that of a co-crystallized complex with 2,3-napthalenediol (PDB: 6FBA) indicated the allosteric mechanism of *Pf*ATCase and the potential for the development of a non-competitive inhibitor [55,56].

From the *Pf*ATCase apo T-state structure to the R-state citrate complex, it is observed that the CP/Asp domains undergo a large conformational change, closing around the bound substrates, leading to a narrowing of the active site [56]. In addition, changes in the electron density of residues near the active site suggested stabilization of these residues by the citrate binding site [56]. In the T state, the CP/Asp domains are relatively more open, resulting in the observed lower substrate affinity and enzymatic activity. This regulatory mechanism allows for precise control and modulation of the enzyme activity, ensuring appropriate pyrimidine biosynthesis in response to cellular demands. Understanding the structural and dynamic differences between the T and R states provides valuable insights into the allosteric regulation of *Pf*ATCase and offers opportunities for the development of allosteric inhibitors that can modulate its activity for potential therapeutic applications.

This situation was further identified through high-throughput activity screening of *Pf*ATCase. The compound 2,3-napthalenediol was confirmed to exhibit inhibition of *Pf*ATCase with a potency in the single-digit micromolar range of $IC_{50}$ [56]. The crystals of the *Pf*ATCase-2,3-napthalenediol complex were grown, and the resulting structure revealed that 2,3-napthalenediol binds near the active site, influencing the shift of one of the key residues, Arg109, towards the active site. This conformational change creates a cavity where 2,3-napthalenediol is accommodated. This mobility of key residues further strengthened the case for future drug design based on allosteric inhibitors of *Pf*ATCase.

## 6. Fragment Screening of *Pf* ATCase

To identify potential starting points, fragment screening was performed using a 140-member subset of an in-house multicomponent reaction (MCR)-compatible library. After establishing the presence of potential hits within this library, an activity assay was then used to narrow down the number of fragments employed in subsequent crystallization experiments, resulting in four fragment hit candidates (Figure 3a–e) (PDB: 7ZCZ, 7ZEA, 7ZGS, and 7ZHI, respectively) (Figure 3a–e) [43]. Experiments were performed by first establishing the DMSO tolerance of the crystal system [63,64]. A series of soaking experiments [65] demonstrated that 5% (*v/v*) DMSO could be tolerated without significant loss of diffraction properties. Concentrations higher than 5% DMSO were found to influence the space group of the *Pf* ATCase crystal. Additionally, the duration of soaking in DMSO was observed to impact the crystals. Extended soaking times in the crystal buffer containing more than 5% DMSO could lead to the dissolution of *Pf* ATCase. The DMSO tolerance of a crystal system is of paramount importance, as the majority of drug-like fragments are weakly soluble or insoluble in aqueous environments [66]. This represents a potential limitation on systems that are amenable to fragment screening. Soaking experiments were carried out from 10 mM stock solutions in 100% DMSO, meaning a maximum final concentration of fragment in each soaking experiment of 0.5 mM. This again represents a major limitation in the identification of even weakly bound fragments, as insufficient fragments are available to saturate binding site(s) and provide unambiguous electron density for the identification of binding poses. No attempt was made to optimize soaking times, which allows the possibility for missed "hits" within the fragments analyzed, but this risk is offset by the size of the library (1020 members). However, weakly bound fragment electron density is typically noisy in appearance and may even be difficult to identify. Fortunately, recent developments in data interpretation [16] allow the identification of even weakly bound fragments through the collection of a large number of "ground state" unbound data sets. Briefly, comparison of the fragment-bound data to a large collection of ground-state data provides a significantly improved model to compare the fragment data too, allowing much weaker binding events to be elucidated. This approach represents a major breakthrough in fragment-based screening, at the cost of a significant increase in synchrotron beamtime and associated computing power. These limitations are offset by the increasing use of robotics and automation in sample preparation, data collection, and data processing, as described briefly in the introduction.

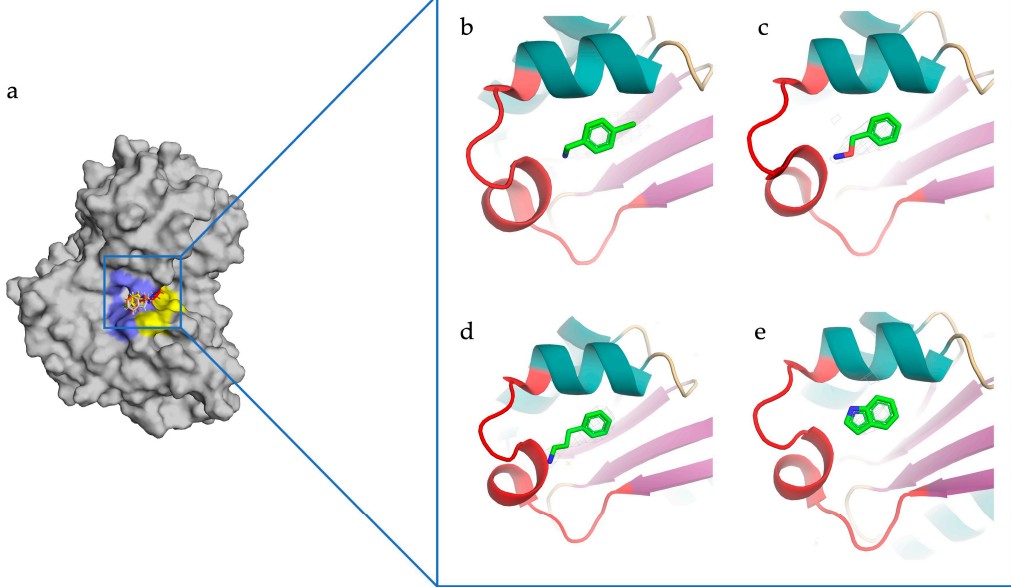

**Figure 3.** *Cont.*

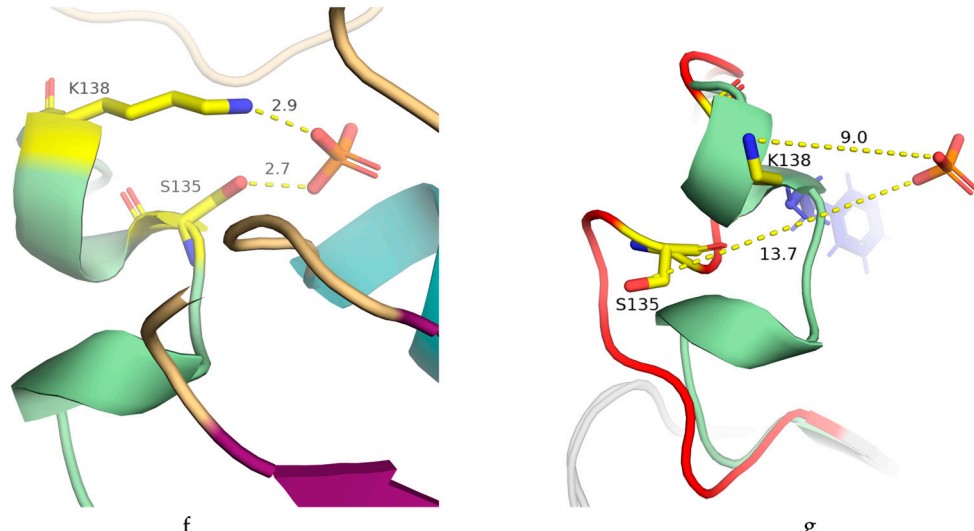

**Figure 3.** The binding mode of fragments to *Pf* ATCase. (**a**) The four fragments bind to the allosteric pocket (blue), which is located in a zone contiguous to the substrate binding pocket (yellow). (**b–e**) the fragments (shown in electron density maps combined with green stick models) -c2b (PDB ID: 7ZCZ), obz (PDB ID: 7ZEA), pra (PDB ID: 7ZGS), ind (PDB ID: 7ZHI)- binding to the allosteric pocket that consists of α-helix and β-sheet elements (the 120 s loop is highlighted in red). (**f,g**) An overlay analysis shows the binding of fragment pra to the allosteric pocket, inducing the conformation change of the 120 s loop from T state (red loop structure) to R state (green helix and loop structure). The key amino acids Ser135 and Lys138 (shown with a yellow stick model) shift 11 Å and 6.1 Å, respectively, with phosphate as a reference point (shown as an orange stick model). This likely contributes to decreasing the flexibility of this region in T-state *Pf* ATCase. Figure created using PyMol v.2.5.5 [58].

In our data, structure alignment analysis reveals a different binding site compared with our previous published citrate complex that mimicked the substrate-*Pf* ATCase binding mode (PDB: 5ILN), located nearby a previously revealed pocket. The new binding pocket consists of two α-helices and a β-sheet, forming a hydrophobic cavity. Intriguingly, these structures provided new binding modes that showed potential for allosteric regulation by changing the conformation of the *Pf* ATCase substrate binding pocket (Figure 3f,g).

## 7. Drug Candidate Design of *Pf* ATCase

Based on the enzymatic assay screening of a 1020-member in-house fragment library and the availability of co-crystals of fragments with *Pf* ATCase, there are five compounds identified to bind to *Pf* ATCase that were found to exhibit weak inhibition in subsequent enzymatic assays [43]. Notably, these five fragments shared a common core structure, a phenylthiophene skeleton, including FLA-01 (PDB: 7ZST). The complex structure of *Pf* ATCase with FLA-01 revealed that the core structure of FLA-01 occupies the new pocket. A polar contact was observed between the Arg109 residue and the methoxy group of FLA-01, providing further evidence of the significance of the Arg109 residue in the active site, as suggested in previous experiments. Additionally, the methoxy side chain is buried within the allosteric binding site and is surrounded by a flexible loop (Arg109-Glu140), which we termed the "120 loop". Moreover, it also forms a polar contact with Thr110, another residue involved in the interaction with the methoxy group [43].

Comparing the structure models of *Pf* ATCase with FLA-01 and additionally obtained fragment structures (fragments A to D), using fragment A as an example, it was observed that the methoxy group partially overlaps with fragment A, but the main portions occupy distinct sites (Figure 4a,b). This suggests that a chemical linkage between these two fragments could be achieved through the methoxy group as a function group and compound FLA-10 as a core structure. Following this approach and with the aim of optimizing inhibition, we further screened various functional groups to replace the methoxy group and

also varied the amino group and aromatic rings. In total, we synthesized 70 compounds in this series, leveraging their compatibility with multicomponent reaction chemistry, and named the resulting library the BDA series. These compounds were screened in an in vitro activity assay, with the best one showing nanomolar (nM) inhibition. The lead compound, BDA-04, underwent co-crystallization, and the resulting complex structure (PDB: 7ZP2) revealed that one side chain, the Boc moiety of BDA-04, partially occupied the CP domain, forming a polar interaction with Arg109 and Ser107 [43]. Additionally, another side-chain benzene group was buried in a hydrophobic area, forming a cation-π interaction with Arg109-Glu140, which is adjacent to the CP domain. We also observed that BDA-04 blocks access to the allosteric pocket through a salt bridge with Arg295, and the "120 loop" appeared to shield this pocket (Figure 4c,d). Another co-crystallization was performed using BDA-14 (PDA:7ZID); this complex structure provided a similar binding pose, but distinctions in the resulting structures provide data for future improvements in this compound. Combining the complex structure with other experiments such as enzyme activity assays, microscale thermophoresis (MST), and Differential scanning fluorimetry (DSF) confirmed the noncompetitive mechanism. Finally, the best-performing candidates were assessed for cytotoxicity against a panel of human cells and their potential to inhibit *P. falciparum* proliferation in a blood-stage malarial culture. Of the current BDA series, BDA-06 shows an $EC_{50}$ of 2.43 μM with good cytotoxicity.

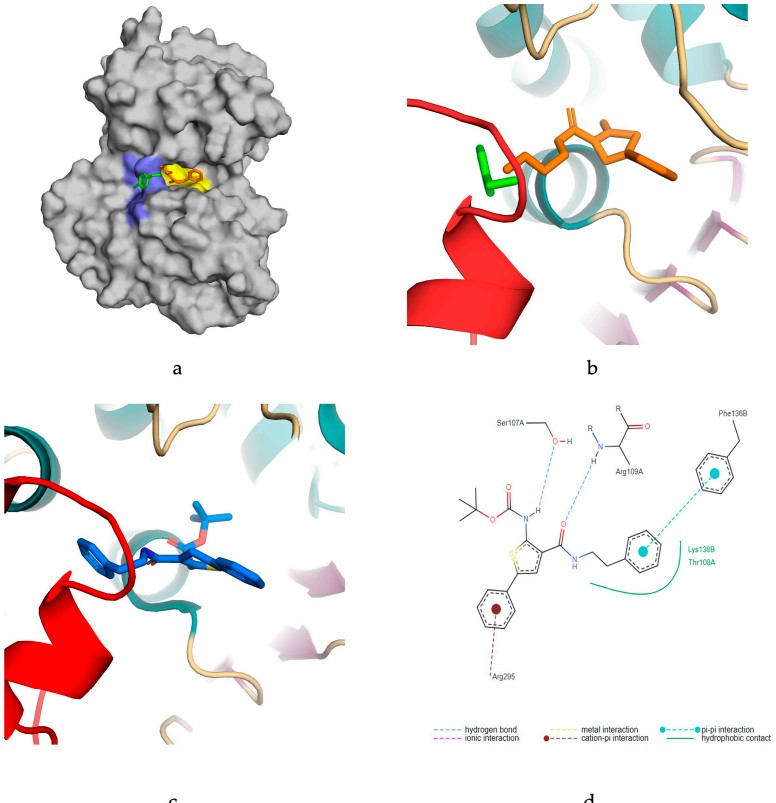

**Figure 4.** (**a**) An overlapped view of 7ZCZ with 7ZST shows fragments A (PDB ID: 7ZCZ) and FLA-01 (PDB ID: 7ZST) located on the allosteric and substrate binding pockets (ligand shown in the stick model), respectively. (**b**) The benzene side chain of compound FLA-01 (shown in the orange stick model) penetrates deep into the substrate binding pocket, and the methoxy side chain extending to the allosteric pocket shows an overlap with fragment A (shown in the green stick model). (**c**) The crystal structure (PDB ID: 7ZP2) of compound BDA-04 (shown as a blue stick model) shows binding to both the allosteric and substrate binding pockets. (**d**) Forming two hydrogen bonds with Ser107 and Arg109; π−π interactions with Phe136; action-π interaction with Arg295; and hydrophobic interaction with Lys138 and Thr108. Figure created using PyMol v.2.5.5 [58].

## 8. Conclusions

The approach of designing and optimizing fragment-based drug candidates, combined with related functional assays, has proven to be a successful strategy for new drug development, particularly in the realm of small-molecule drug discovery [67]. This approach typically begins with high-throughput screening of fragments, aiming to identify efficient fragments even with very low affinity. Subsequently, these identified fragments are used as starting points for the design of target inhibitors or constructing the core structure of new ligands. Throughout these stages, additional techniques, such as biophysical binding affinity tests and assessments of inhibition stability, are also employed. With this strategy, the utilization of crystal structure is instrumental in elucidating intricate protein details and mechanisms. This review chronicles the progression through the stages of discovering an allosteric inhibitor of the malarial aspartate transcarbamoylase (ATCase), showcasing the initial and intermediate phases of developing new inhibitors through the fragment-based approach.

A key contribution to the success of fragment-based screening was the design decision to screen fragments that were directly amenable to multicomponent reaction chemistry. While alternative libraries exist, a frequently encountered issue is the complexity of synthesizing "follow-up libraries" that allow researchers to explore the chemical space proximal to the hits they have determined. Under a linear chemical synthesis approach, this provides severe limitations on the size of such a follow-up library and an associated increase in the number of optimization cycles that may be required. A significant contributor to this limitation is the need to purify each stage of a linear reaction. The core features of multicomponent reaction chemistry are both the ability to create a novel molecule around a central scaffold in a single reaction and the ability to vary the substituents around this core scaffold in a combinatorial manner. Thus, a 3-component reaction using 125 substituent variants for each component (as performed in this example) would yield $5^3$ or 125 distinct molecules. This provides a library of highly similar molecules suitable to provide detailed feedback on the impact of substituent variation on binding, inhibitory efficacy (in this example), and binding modes. In this example, all these molecules would be available from only 15 starting components in reactions that can be run in parallel, and each would require only a single purification step. While differences in synthesis yields will play a role, sufficient samples to provide experiment data for good coverage of the local chemical space can reasonably be expected (70 in the case of the BDA series).

Apart from generating a library of compounds targeting an allosteric site of *Pf* ATCase, this library has high potential for probing similar sites of other organisms previously shown to be sensitive to inhibition of de novo pyrimidine biosynthesis, including human, *Mycobacterium tuberculosis*, *Trypanosoma cruzi*, and *Arabidopsis thaliana* ATCases. The relative lack of sequence conservation (when compared to the absolutely conserved active site residues in these organisms) may yet provide a class of molecules that can be broadly applied to address a number of different diseases or biotechnological problems. This diversity underscores the potential for exploitation of allosteric sites, in the discovery of which fragment-based screening by X-ray crystallography is likely to remain an important tool in the future.

The strategy of fragment-based drug design has proven to be a promising approach in drug discovery for decades. It is target-driven, focusing on interactions with the biological target of interest. This process typically begins with the screening of small and simple fragments, allowing for more efficient hit identification and easier optimization of interactions [68]. Through an iterative optimization process, there is an opportunity to improve binding affinity and other drug-like properties, thereby enhancing the likelihood of success in developing lead compounds. Coupled with X-ray crystallography [69] or NMR spectroscopy [70], this approach can help identify the interactions between the target protein and small fragments, guiding decision-making about the feasibility of designing optimized ligands. It is worth noting that this process significantly relies on the successful outcome of several independent techniques. In the preparation stage before fragment-based

screening, target protein sequence identification, protein purification, and the achievement of an active protein are required. For example, the Human Genome Project (HGP) [71], combined with modern protein purification techniques [72], has facilitated access to purified active target proteins. Moreover, HTS methods [25] effectively handle large libraries of fragments. However, changes such as a low hit rate in lead fragment screening, potential optimization challenges, limited initial affinity, solubility of fragments, and optimized candidates cannot be ignored, and the consensus of the "correct" fragment library is as yet unformed. In addition, another challenge pertains to crystallization, as not all proteins are able to form crystals. Even if the apo crystal structure is obtained, it does not guarantee the success of obtaining a co-crystal. Although virtual structure models provide the possibility of serving as protein target structures [73], the difference in results between real crystals and virtual crystals, along with the complexity of the real environment of crystal growth, cannot be ignored.

## 9. Patents

The authors have submitted a patent application on the molecules described in this manuscript (No. 21211903.6).

**Author Contributions:** Conceptualization, X.D., R.Z. and M.R.G.; writing—original draft preparation, X.D. and R.Z. All authors have read and agreed to the published version of the manuscript.

**Funding:** X.D. and R.Z. would like to thank the Chinese Scholarship Council for funding.

**Data Availability Statement:** All structures described in this manuscript are deposited with the RCSB under the indicated accession numbers.

**Conflicts of Interest:** The authors declare no conflict of interest.

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
