# Peer review of "Fragment Screening in the Development of a Novel Anti-Malarial"

_crystals, doi:10.3390/cryst13121610_

Round 1

Reviewer 1 Report

Comments and Suggestions for Authors

Review for manuscript crystals-2680709 :

Fragment Screening in the Development of a Novel Anti-malarial

X. Du et al. are presenting a review on their work developing novel anti-malarials assisted by fragment screening. Fragment screening has recently become mature technique and campaigns are support by most synchrotron light sources. Unfortunately, the authors miss the opportunity to review these developments, in particular the efforts towards drugs against SARS-Cov-2. The manuscript isn’t written well and for a review important developments are omitted. Only after reading the 2022 JACS paper from the same authors (citation 47) I was able to understand this review. In fact the presented manuscript almost exclusively reviews the 2022 paper and by doing this even lacks some of the important details (first 140 small fragments, then 1020 fragments with activity assay based selections). I personally see no additional value in this review. Hence, I do not recommend this manuscript for publication. 

A few but not the complete list of my main concerns:

1. Introduction

Crystallisation is not mentioned at all, also high-throughput approaches for soaking and crystal fishing are overlooked. While remote operation of synchrotron beamlines has been standard for some time, many beamlines allow fully automated un-attended data collections.

The list of software developments is incomplete, at least Dials and Phenix should be mentioned. While PanDDA is mentioned later as a major  breakthrough in section 6, this should be covered here.

As mentioned above several fragment screening campaigns were run at several synchrotrons and should be mentioned.

Explanation of what a BDA series is only given in section 8.

2. Pyrimidine synthesis pathway

I had to read other sources to understand this.

6. Fragment screening & 7. Drug candidate design

The description of the fragment libraries is missing. Only reading the 2022 JACS paper reveals the fact that two libraries were used. With an activity assay based selection process used to narrow down the number of fragments used in crystallisation experiments. The section discusses the issue of DMSO tolerance (no reference to this well known limitation) and the potential that soaking times might have limited the success rate. Figure 2 shows that the 4 fragments from the first screen bind in a similar way sharing a phenyl ring, no electron density maps are shown. 

Comments on the Quality of English Language

Some sentences are too long and the use of past and presence tense should be checked.

Author Response

Dear editor,

Thank you very much for giving us the opportunity to revise our manuscript entitled ‘Fragment Screening in the Development of a Novel Anti-malarial’. We are also particularly grateful to reviewers for their constructive comments and suggestions. There comments have stimulated us to go back over the manuscript and make substantial changes. We marked the changes in the manuscript in yellow. We sincerely hope that the revisions we have made meet your expectations.

Our response to the review is provide point-by-point below:

Referee 1

  1. Du et al. are presenting a review on their work developing novel anti-malarials assisted by fragment screening. Fragment screening has recently become mature technique and campaigns are support by most synchrotron light sources. Unfortunately, the authors miss the opportunity to review these developments, in particular the efforts towards drugs against SARS-Cov-2. The manuscript isn’t written well and for a review important developments are omitted. Only after reading the 2022 JACS paper from the same authors (citation 47) I was able to understand this review. In fact the presented manuscript almost exclusively reviews the 2022 paper and by doing this even lacks some of the important details (first 140 small fragments, then 1020 fragments with activity assay based selections). I personally see no additional value in this review. Hence, I do not recommend this manuscript for publication. 

Response: Thanks for your kindly comments. We also appreciate that the subject matter is large, and has also been covered by other reviews of fragment screening by X-ray or other techniques frequently in the literature. Our intention in this review was to provide and describe data that we had insufficient space for in our JACS manuscript and use this as an example of specific pitfalls we faced and addressed during our successful fragment-based inhibitor discovery.

While the aspect of the initial 140-member fragment library screened by Xray and the 1020-member fragment library screened by a cocktail-based assay were mentioned in introduction section (page 3 > introduction > fourth paragraph > line 87-89, reference 43) we have clarified the text to make it clearer to the reader.

  1. Introduction

Crystallisation is not mentioned at all, also high-throughput approaches for soaking and crystal fishing are overlooked. While remote operation of synchrotron beamlines has been standard for some time, many beamlines allow fully automated un-attended data collections.

Response: Automation of crystal data collection as well as synchrotron beamlines are now more clearly introduced in the introduction section (page 3 > introduction > second paragraph > line 46-53, references 18-21); High-throughput approaches for soaking in the introduction section (page 2 > introduction > second paragraph > line 58-63, references 24-26) and crystal fishing (page 2 > introduction > second paragraph > line 63-66, references 27-29).

The list of software developments is incomplete, at least Dials and Phenix should be mentioned. While PanDDA is mentioned later as a major  breakthrough in section 6, this should be covered here.

Response: Thanks for your advice, since PanDDA is a method pipeline rather than a stand alone software, the ‘software developments’ section was changed to ‘data processing developments’ to include this method, and include a description of Dials and Phenix (Page 2 > Introduction > first paragraph > line 37-38, references 13, 14 and 16).

As mentioned above several fragment screening campaigns were run at several synchrotrons and should be mentioned.

Response: the synchrotrons used for data collection are now indicated in the introduction section (page 3 > introduction > fourth paragraph > line 89-90, reference 43).

Explanation of what a BDA series is only given in section 8.

Response: the introduction of the BDA series is now explained in the introduction section (page 3 > introduction > fourth paragraph > line 95).

  1. Pyrimidine synthesis pathway

Response: The de novo pyrimidine biosynthesis pathway is further detailed in ‘2. pyrimidine biosynthesis as a malarial target’ section and figure 1 (page 3 > 2. pyrimidine biosynthesis as a malarial target > second paragraph > line 120-125; reference 49; page 9 > figure 1).

  1. Fragment screening & 7. Drug candidate design

The description of the fragment libraries is missing. Only reading the 2022 JACS paper reveals the fact that two libraries were used. With an activity assay based selection process used to narrow down the number of fragments used in crystallisation experiments. The section discusses the issue of DMSO tolerance (no reference to this well known limitation) and the potential that soaking times might have limited the success rate. Figure 2 shows that the 4 fragments from the first screen bind in a similar way sharing a phenyl ring, no electron density maps are shown. 

Response: the description of the fragment library is now clarified  in ‘6. Fragment screening of PfATCase’ and ’7. Drug candidate design’ (page 6 > 6. Fragment screening of PfATCase > first paragraph > line 250-251; page 7 > 7. Drug candidate design > first paragraph > line 289, reference 43); a discussion of DMSO tolerance is included in ‘6. Fragment screening of PfATCase’ with references 62, 63, and soaking with reference 64 (page 6 > 6. Fragment screening of PfATCase > first paragraph > line 254-260, reference 62 and 63); the requested electron density maps were added to figure 3 (Page 11 > figure 3).

Reviewer 2 Report

Comments and Suggestions for Authors

Authors nicely summarize their research to develop a novel anti-malarial. ATCase is involved in de novo pyrimidine nucleotide biosynthesis and is considered as a drug target for pathogenic parasites such as malaria-causing protozoan because they lack salvage pathway for pyrimidine nucleotide biosynthesis. Authors determined the crystal structure of PfATCase and searched lead compounds using fragment screening method. They found two types of compounds; one is bound to the active site, and the other is bound to the allosteric site nearby the active site. They chemically synthesize a new compound in which two types of fragments are linked to improve affinity and determined the structure in complex with a novel compound. They further evaluate the biochemical characters and toxicity of a novel type of compound. This is a good example to develop a druggable compound by fragment screening, thus I recommend publishing the manuscript in Crystals with minor revisions as shown in below comments.

Minor comments

The models of ligands in Figures 2 and 3 contain hydrogen atoms although these positions are not experimentally characterized. I recommend to mention it in figure legends.

Overlapped views are represented in figures 2f, 2g, and 3b, but the use of different colors is not explained. For example, green helix and loop structure shown in 2f and 2g represents the ligand free structure of 120s loop, but it is not explained in figure legend. Fragment A and FLA-01 are green and orange models, respectively, but not explained in the legend. Please explain more details.

Chain names are included in the residue name in Figure 3d, but only the chain name of Arg295 is missing.

Typos

P5, line 196, The active sit of E.coli -> The active site of E.coli

Author Response

Authors nicely summarize their research to develop a novel anti-malarial. ATCase is involved in de novo pyrimidine nucleotide biosynthesis and is considered as a drug target for pathogenic parasites such as malaria-causing protozoan because they lack salvage pathway for pyrimidine nucleotide biosynthesis. Authors determined the crystal structure of PfATCase and searched lead compounds using fragment screening method. They found two types of compounds; one is bound to the active site, and the other is bound to the allosteric site nearby the active site. They chemically synthesize a new compound in which two types of fragments are linked to improve affinity and determined the structure in complex with a novel compound. They further evaluate the biochemical characters and toxicity of a novel type of compound. This is a good example to develop a druggable compound by fragment screening, thus I recommend publishing the manuscript in Crystals with minor revisions as shown in below comments.

Minor comments

The models of ligands in Figures 2 and 3 contain hydrogen atoms although these positions are not experimentally characterized. I recommend to mention it in figure legends.

Response: the hydrogen atoms are removed from these two figures (page 11 > figure 3; page 12 > figure 4).

Overlapped views are represented in figures 2f, 2g, and 3b, but the use of different colors is not explained. For example, green helix and loop structure shown in 2f and 2g represents the ligand free structure of 120s loop, but it is not explained in figure legend. Fragment A and FLA-01 are green and orange models, respectively, but not explained in the legend. Please explain more details.

 Response: the different colors are now explained in the associated figure legend (page 11 > figure 3 > legend highlight in yellow; page 12 > figure 4 > legend highlighted in yellow).

Chain names are included in the residue name in Figure 3d, but only the chain name of Arg295 is missing.

Response: Arg295 is now shown in figure 4d (page 12 > figure 4d).

Typos

P5, line 196, The active sit of E.coli -> The active site of E.coli

Response: The active sit of E.coli in line 196 was changed (page 5 > line 216).

Reviewer 3 Report

Comments and Suggestions for Authors

The manuscript by Du et al. set a focus on fragment-based screening of novel antimalarial agents. The topic is interesting and the manuscript is mainly well-written, but its focus is very narrow since it reads like a description of own work rather than give an overview on the topic/field. This manuscript is publishable, but it needs to include some general aspects of the topic.
What are key challenges in fragment-based design and what are common strategies to address them? What are advantages and limitations of this approach compared to others? These points might be addressed in a separate paragraph, or throughout the manuscript.

Comments on the Quality of English Language

Please read and check the manuscript carefully. It is in general well-written, but some phrases need to be corrected.
e.g. page2, line 69 'Fragment-based screening through X-ray...' should be 'Fragment-based screening through X-ray crystallography'

Author Response

The manuscript by Du et al. set a focus on fragment-based screening of novel antimalarial agents. The topic is interesting and the manuscript is mainly well-written, but its focus is very narrow since it reads like a description of own work rather than give an overview on the topic/field. This manuscript is publishable, but it needs to include some general aspects of the topic.

What are key challenges in fragment-based design and what are common strategies to address them? What are advantages and limitations of this approach compared to others? These points might be addressed in a separate paragraph, or throughout the manuscript.

Response: Many thanks for this suggestion. These points are now addressed in a separate paragraph in ‘8. conclusion section’ (page 8 > 8. conclusion section > last paragraph > line 373-395, reference 67-72).